# Influence of Intact Mycelium of Arbuscular Mycorrhizal Fungi on Soil Microbiome Functional Profile in Wheat under Mn Stress

**DOI:** 10.3390/plants11192598

**Published:** 2022-10-02

**Authors:** Taiana A. Conceição, Galdino Andrade, Isabel Brito

**Affiliations:** 1Federal University of Recôncavo of Bahia, Bahia 44574-490, Brazil; 2MED—Mediterranean Institute for Agriculture, Environment and Development & CHANGE—Global Change and Sustainability Institute, Institute for Advanced Studies and Research, University of Évora, Pólo da Mitra, 7006-554 Évora, Portugal; 3Department of Microbiology, State University of Londrina, Paraná 86051-990, Brazil

**Keywords:** wheat bio-protection, arbuscular mycorrhiza, Mn toxicity, soil biological activity, metabolic soil profiling

## Abstract

In the current agronomic context, the adoption of alternative forms of soil management is essential to increase crop yield. Agricultural sustainability requires practices that generate positive impacts and promote an increase in microbiome diversity as a tool to overcome adverse environmental conditions. An important ally is the indigenous arbuscular mycorrhizal fungi (AMF) that can improve plant growth and provide protection against abiotic stress such as metal toxicity. In a greenhouse experiment, this work studied the effect of wheat growth on several parameters of biological activity and functional microbiome in relation to wheat antecedent plant mycotrophy and soil disturbance under Mn stress. When the wheat was planted after highly mycotrophic plants and the soil was not previously disturbed, the results showed a 60% increase in wheat arbuscular colonization and a 2.5-fold increase in dry weight along with higher values of photosynthetic parameters and dehydrogenase activity. Conversely, soil disturbance before wheat planting increased the β-glucosidase activity and the count of manganese oxidizers, irrespectively of antecedent plant, and decreased drastically the wheat dry weight, the AMF colonization and the chlorophyll content compared to the undisturbed treatment. These findings suggest that not only the wheat growth but also the soil functional microbiome associated is affected by the antecedent type of plant and previous soil disturbance imposed. In addition, the improvement in wheat dry weight despite Mn toxicity may rely on shifts in biological activity associated to a well-established and intact ERM early developed in the soil.

## 1. Introduction

Over the past few decades, the great challenges of agriculture systems to overcome the problem of food production are how to increase the crop yield without increasing the area of harvested land. Agriculture management with minimum soil disturbance and crop diversification has been linked with higher soil quality and crop yield. This conservation management approach affects soil microorganisms and creates a favorable environment that will affect plant nutrition and may protect crops against abiotic stress [1,2]. Several strategies for the effective exploitation have been proposed to optimize the role of the root-associated microbiome in nutrient supply and plant protection [3].

Extensive areas of soil in the south region of Portugal are characterized by acidic properties that promote an increase in Mn ions bioavailability and cause great toxicity to the crops [4]. The excessive Mn availability can impact the plant growth by affecting chlorophyll biosynthesis, replacing cofactors in enzymes involved in the photosynthetic pathway and interfering in early reactions in photosystem II (PSII), causing a decline in photosynthetic rate and plant development [5]. Mycorrhizal associations can be related to several benefits to host plants, not only improving nutritional state but also helping the crop to overcome metal stress, particularly in well-established colonizations [6,7,8]. Among the different forms of arbuscular mycorrhizal fungi (AMF) inoculum, the intact extra radicular mycelium (ERM) results in an earlier and fast root colonization and therefore leads to a tolerance to metal stress [9].

A bio-protection strategy was developed by Brito et al. [6] to overcome the problem of Mn toxicity in acidic soils by introducing a mycotrophic antecedent plant to develop an ERM that, when not disrupted, promotes an early and effective wheat root colonization. In that study, it was reported that the wheat that grew after the mycotrophic plants, *Ornithopus compressus* (legume) and *Lolium rigidum* (grass), with an intact ERM reduced the wheat shoot Mn content by 47% and 36%, respectively, doubled the shoot phosphorus (P) content and led to a 1.5-fold increase in wheat dry weight compared to the same mycotrophic plants with ERM disrupted. This approach takes advantage of an early root colonization by well-adapted indigenous AMF developed by Mn-tolerant plant species present in natural vegetation to promote bio-protection in the subsequent crop. Mycorrhizal associations affect not only plant development but also act as determinants of the microorganism’s community dynamics. The large surface area of the ERM provides nutrient-rich niches for the colonization and growth of other soil microorganisms, especially bacteria, and it seems to have a specific selection pressure on the microbe composition [10]. In fact, the differences observed in the bio-protection strategy are accrued to the functional microbiome shaped by the AMF diversity managed by each antecedent plant [11].

The changes in wheat functional microbiome induced by the antecedent plant can be assessed by measuring the key microbial and biochemical processes, in addition to other biological attributes, to evaluate the influence of crop practices on soil function [12]. Soil microbial biomass carbon has been commonly recognized as an important indicator of soil microbial parameters. It represents the size of the microbial pool, which reflects soil organic matter (SOM) changes such as carbon cycling [13]. Soil respiration is also closely related to several functions of organisms. The measurement of soil basal respiration, which originates from the mineralization of SOM, has been applied across a variety of studies and could be used to assess changes imposed by agricultural practices [14].

Soil enzymatic activities are also considered useful indicators of soil status because of their involvement in the decomposition of SOM and rapid response to changes in soil management. Major groups of enzymes are used to evaluate soil status. Dehydrogenases are an oxidoreductase group correlated to the activity of viable cells. Arylsulfatase, phosphatase and β-glucosidase are hydrolases involved in the cycling of sulfur, phosphorus and carbon, respectively, from organic composts [15]. Due to substrate specificity, a group of enzymatic activities is necessary to infer the general status of the soil or microbiological activity indices. However, there may not be a simple relation between the measurement of enzymatic activity and the microbial functional diversity, hence the complexity of metabolic reactions and interactions of soil microbiota [16]. Since relationships between soil enzymes and other parameters of soil biological activity are not direct, they need to be analyzed carefully [13].

Although bacterial count is considered a laborious methodology, it can still be used as an initial point of study due to its relatively inexpensive cost and also by assessing the gross functional diversity of culturable microorganisms [17,18]. Therefore, since heterotrophic microorganisms are dominant drivers of biogeochemical cycles, shifts in the functional microbe count could be used as a preliminary evaluation of the agronomic management impact, and several studies have demonstrated the impact of agricultural practices on soil microbial count [19,20,21].

Studies that link microbial communities’ response to different agricultural practices and perturbations could help predict the outcome of specific management intervention and maximize the sustainability of soil resources [22]. Therefore, this work aims to build up the knowledge involved in the strategy of AMF bio-protection against Mn toxicity in wheat growth proposed by Brito et al. [6], studying the changes in soil functional profile induced by wheat growth in relation with antecedent plant mycotrophy and soil disturbance from the perspective of soil biological activity and microbial functional response.

## 2. Results

### 2.1. Wheat Root Colonization, Photosynthetic Parameters, and Shoot Development

Wheat plants that grew after the highly mycotrophic plants (*O. compressus* and *L. rigidum*) reported the greatest means of root colonization by arbuscular mycorrhizal fungi (AMF), particularly in undisturbed soil (Figure 1). In the disturbed treatment, the mycotrophy of the antecedent plant does not seem to be relevant for arbuscular colonization, as no statistical differences were observed between plants. The same pattern of results was observed in relation to wheat shoot dry weight (DW). Although to a lower extent, soil disturbance led to a decrease in the wheat shoot dry weight also when grown after the non-mycotrophic plant (Figure 2).

For photosynthetic parameters (Figure 3), wheat that grew after mycotrophic plants showed a higher chlorophyll content in the undisturbed treatment. Soil disturbance did not affect the electron transport chain rate (ETR), photochemical quenching (qP) and stomatal conductance to water vapor (gs) after mycotrophic plants, denoting a greater resistance to physiological damage in photosynthetic apparatus. Soil disturbance only led to a great decrease in these parameters in wheat that grew after the non-mycotrophic plant. No significant differences between treatments were observed for the photosynthetic rate (A) and maximum quantum efficiency of PSII (Fv/Fm), but globally, the wheat that grew after no previous plant exhibited the lower A and Fv/Fm (Table 1).

### 2.2. Soil Microbial Activity

Even though soil microbial biomass carbon (C-Mic) after wheat growth was not affected by the mycotrophy of the antecedent plant nor soil disturbance (Table 2), when wheat grew with no previous plants, lower values of basal respiration (SBR) and consequently lower metabolic quotient (qCO_2_) was observed. The soil where wheat grew after the legume (*O. compressus*) presented the highest mean of SBR. There were no significant differences between the undisturbed and disturbed treatment regarding the general biological activity measured by C-Mic, SBR and qCO_2_ parameters.

### 2.3. Functional Groups of Culturable Microorganisms

The antecedent plant affected the mean count of total bacteria, Mn oxidizers and P solubilizers after wheat growth, but not related to plant mycotrophy (Table 3). For P solubilizers, the greatest mean was found in the soil in which wheat grew after the highly mycotrophic *L. rigidum*, and the lowest mean after the non-mycotrophic *S. gallica*. The soil disturbance affected the mean of total bacteria, Mn oxidizers, ammonifiers and S oxidizers, causing an increase in these microbial counts when compared with the undisturbed treatment. The Mn oxidizers count is higher irrespective of the wheat antecedent plant in the disturbed soil treatment (Figure 4), but in undisturbed treatment, significant differences can be identified between the mycotrophic *L. rigidum* and the non-mycotrophic *S. gallica* as antecedent plants, with a higher value for the latter. 

### 2.4. Enzymatic Activity

Considering the antecedent plant mycotrophy, the higher enzymatic activities were observed in the soil where wheat grew after the mycotrophic plants except for phosphatase (Table 4). Previous soil disturbance affected the mean of dehydrogenase and the β-glucosidase activity, by strongly decreasing the first (about 60%) and increasing the last, after wheat growth. For dehydrogenase activity, there was a significant interaction between the type of antecedent plant and soil integrity (Figure 5). The decreased of dehydrogenase activity after soil disturbance was significant only for wheat that grew after mycotrophic plants. The greatest dehydrogenase activity was observed in the soil where wheat grew after *L. rigidum* in the undisturbed treatment.

### 2.5. Correlation Analysis

In the undisturbed treatment, the Pearson correlation test showed a strong correlation coefficient of root colonization by arbuscular mycorrhizal fungi (AMF) with dehydrogenase activity (0.636), soil basal respiration (0.776), total bacteria (−0.608), Mn oxidizers (−0.517), dry weight (0,853), chlorophyl content (0.594), maximum quantum efficiency of PSII (0.599) and photochemical quenching (0.754). The linear regression analysis showed a great coefficient of linear regression between root colonization by AMF and shoot dry weight (DW) and soil basal respiration (SBR), and it showed a moderate coefficient of linear regression between root colonization by AMF and photosynthetic parameters (Figure 6).

In the disturbed treatment, a correlation between root colonization by AMF and β-glucosidase activity (0.783), total bacteria (−0.559) and photosynthetic rate (−0.580) was found according to the Pearson correlation test. In the linear regression (Figure 7), a moderate coefficient was found between root colonization by AMF and β-glucosidase (B-GLIC) activity and weak coefficient was found in relation with total bacteria (Bac) and photosynthetic rate (A).

## 3. Discussion

This experiment used a bi-factorial design to evaluate the effect of wheat growth under Mn toxicity on soil microbial activity and functional diversity, taking into consideration the influence of antecedent plant mycotrophy and soil disturbance. It was observed that the antecedent plant mycotrophy and keeping the soil undisturbed gave rise to a significant improvement in wheat dry weight and AMF colonization. The extra radicular mycelium (ERM) formed in the soil when kept intact (the undisturbed treatment) allows an early and faster AM colonization, granting wheat protection against Mn stress. This is in agreement with the results found by Brito et al. [6]. Although they found that the AMF colonization rate was significantly higher in wheat that grew after the legume (*O. compressus*) than after the grass (*L. rigidum*), our results showed no statistical differences between these two antecedent plants. We believe that could be attributed to the time that the antecedent plant grew before wheat was sown, 11 weeks in our study instead of 7 weeks in theirs. 

General soil microbial activity (DHA and SBR) when wheat grew after mycotrophic plants in the undisturbed is positively correlated with AMF colonization rate and illustrates the importance of ERM integrity formed by the antecedent plant as an active niche for the survival of soil microbes. The greater microbial activity observed under these circumstances was not accompanied by an increase in the count of total bacteria, phosphorus solubilizers, ammonifiers, or sulfur oxidizers, indicating the presence of unculturable or recalcitrant organisms to the generalist culture media used [23].

The Mn toxicity alleviation, reflected in the shoot dry weight of wheat grown after mycotrophic plants in undisturbed soil, is related to a greater photosynthetic performance (Chl, FV/Fm and qP). Under similar circumstances, Brito et al. [6] observed a Mn shoot reduction in wheat of about 21% after *S. gallica*, 36% after *L. rigidum* and 47% after *O. compressus*, also together with an increase in wheat shoot P content. The toxic effects of Mn affecting photosynthetic parameters have been reported [24,25,26]. Mn toxicity is associated with photosynthetic enzymes alterations that can affect the biosynthesis of chlorophyll. Moreover, total chlorophyll content emerges as an efficient physiological indicator of the functional microbiome induced by the previous plants in undisturbed soil. This effect is clearly observed in the photosynthetic parameters measured in this study. Wheat that grew after non-mycotrophic plants or no previous plants exhibited lower values of electron transfer rate (ETR), photochemical quenching (qP) and chlorophyll content.

High concentrations of Mn in soil can also induce oxidative stress response, indirectly decreasing the photosynthetic activity [27]. The generated reactive oxygen species (ROS) caused by the Mn excess are responsible for damages in photosystem II (PSII) [28]. Changes in qP likely influence ETR and PSII yield [29]. In a similar experiment, Faria et al. [30] observed that AMF symbiosis can induce biochemical alterations that helped wheat counteract metal stress by reducing Mn ion uptake, altering the subcellular Mn allocation and increasing the activity of enzymes involved in stress response. Additionally, the soil disturbance and consequently ERM disruption also led to a great decrease in these parameters after the mycotrophic plants, confirming the importance of an intact ERM and early AMF colonization in Mn toxicity alleviation through the protection of the plant photosynthetic system.

In disturbed soil, the root colonization by AMF has a positive correlation with β-glucosidase activity. The increase in β-glucosidase activity can solely be a result of soil disturbance [31], but the higher activity correlated with root colonization by AMF indicates that the ERM disturbance could also influence the soil enzymatic activity related to C metabolism. It is known that soil disturbance causes a rapid loss in soil organic matter (SOM) content and therefore could reduce crop productivity [32]. In this study, the increase in SOM mineralization ten days after soil disturbance is supported by the lower dehydrogenase activity. In addition, the short-term effects of soil mobilization are generally related to changes in extracellular soil enzyme activity [33], and an increase in β-glucosidase activity has been reported after soil disturbance induced by conventional soil management [31], which is confirmed by our results. The increased aeration associated to soil disturbance also favored an increase in the functional groups of culturable microorganisms and total bacteria. Despite the increased microbial activity and organic matter mineralization, plants failed to cope with the Mn stress when AMF colonization was primary initiated by spores and colonized roots fragments and therefore occurred in a later stage of wheat development. Some nutrient loss due to the rapid mineralization of SOM may have also occurred, as even after the non-mycotrophic *S. gallica*, wheat had a poor growth in disturbed soil. However, the magnitude of the difference in SOM mineralization of the non-mycotrophic is smaller when compared to the mycotrophic antecedent plants, particularly *O. compressus*.

A higher phosphatase activity was found in the soil where wheat grew after the non-mycotrophic plant. This may be accrued to a synergistic activity of this enzyme released by the previous plant root (and accumulated in soil matrix) and its specific microbiome. Since the non-mycotrophic plant does not count with AMF contribution for P acquisition, a deficiency in P uptake could stimulate the release of this enzyme by its specific microbiome [34]. This pattern was not followed by the P solubilizers count that was favored by the mycotrophic antecedent plants, indicating their possible association with the ERM.

Microorganisms that oxidize Mn are linked to a decrease in its availability to plants [35]. The damage to the ERM caused by the disturbed treatment and consequent lower AMF colonization of wheat seems to activate other mechanisms of Mn alleviation by stimulating this functional group as an attempt to mitigate the Mn toxicity. The increase in Mn oxidizers count observed after soil disturbance could be linked to reduced Mn toxicity by immobilizing this bioavailable ion into oxides [36] and aiding the wheat growth under this circumstance. However, Mn concentration of wheat plants grown after *L. rigidum* in disturbed soil was 1.6 times higher (Faria et al. unpublished data) and shoot dry weight was 3 times lower than in undisturbed soil. Our results indicate that changes in the wheat microbiome concerning Mn oxidizers are not associated with the detoxifying mechanisms of bio-protection when wheat grew after a mycotrophic plant in disturbed soil. In contrast, the high Mn oxidizers count is associated to an increase in wheat Mn shoot content and antioxidant enzymatic activity as found by Faria et al. [30] using the same experimental design.

Soil disturbance and the associated disruption of ERM and accompanying microbiome could lead to a loss of microbe interactions, shift the microbe functional complementarity and prevalence of different groups and thus lead to a loss of redundancy [37,38]. Further research is required to study the differences in the rhizospheric microbiome of these plants.

## 4. Materials and Methods

### 4.1. Experimental Design

A pot experiment was performed under controlled conditions from January to May 2019 in a greenhouse. A sandy acidic soil [4] was collected from the first 20 cm of a natural pasture at Herdade da Mitra, University of Évora, Alentejo, Portugal (38°32′ N; 08°00′ W). This soil was characterized by a high AMF diversity [39] and was used in previous experiments [6,7]. To guarantee initial identical conditions in all treatments, the soil was homogenized by sieving and packed into 8 kg pots. The experiment consisted of two phases. In the first one, three common arable plants species, widespread in areas exhibiting soil Mn toxicity, were sown in 8 replicate pots, with 5 plants per pot to develop different levels of extra radicular mycelium. Two species, *Ornithopus compressus* L. (a legume) and *Lolium rigidum* Gaudin (a grass), are known to be highly mycotrophic; and the last, *Silene gallica* L., is non-mycotrophic. After 11 weeks of growth, plants were excised. For the disturbed treatment, the soil of half of the pots for each species was subjected to mechanical disturbance by passing through a 4 mm sieve. The soil and roots were mixed, repacked into the same pots and shoot material was returned to the soil surface. The remainder of the pots of each species formed the undisturbed treatment; the shoot material was also returned to the soil surface. Ten days later, in the second phase of the experiment, wheat (*Triticum aestivum* L., var. Ardila) pre-germinated seeds were planted in all the 24 pots from phase one plus 4 additional pots that did not received any plants in phase 1 and were allowed to grow for 21 days. After that period, wheat photosynthetic parameters, shoot dry weight, mycorrhizal colonization and soil biological activity from all replicates with wheat were measured. Wheat root material was collected and their colonization with AMF was determined after staining with trypan blue, according to the magnified intersections method [40].

### 4.2. Soil Microbial Activity

Water-holding capacity and water content were determined [41], and the information was used to calculate the following parameters. Soil basal respiration (SBR) was measured accordingly to Silva et al. [42] in a closed jar and incubated for 7 days at 26 °C. The CO_2_ released was adsorbed in NaOH and determined by HCl titration. The results are reported as milligrams of CO_2_ per kilograms of soil released per hour. The determination of total microbial biomass carbon (C-Mic) was performed by fumigating the soil with chloroform in a desiccator, and the carbon content was calculated following an oxidation reaction with potassium permanganate [43]. The values of C-Mic are given by the carbon content of fumigated soil minus that of the non-fumigated soils, which were all divided by the proportion of microbial C evolved (kc). A value of 0.45 was used for kc in C-Mic calculation [44]. Results are expressed as milligrams of carbon per kilograms of soil. The metabolic quotient (qCO_2_) was used to estimate the efficiency of substrate consuming by microorganisms as a stress indicator and was calculated as the ratio between soil basal respiration and carbon microbial biomass [45].

### 4.3. Functional Groups of Culturable Microorganisms

Five functional culturable groups of soil microorganisms were evaluated: total bacteria, ammonifiers, sulfur oxidizers, manganese (Mn) oxidizers and phosphorus (P) solubilizers. For bacteria, ammonifiers and P solubilizers, the protocols used were the ones described in Albino and Andrade [46]. Manganese oxidizers were counted in Garretesen’s media as suggested by Nogueira et al. [47]. Sulfur oxidizers were counted in thiosulfate broth [48] using bromothymol blue as an indicator of pH reducing instead of bromocresol purple. Ammonifiers and sulfur oxidizers are presented as the logarithm of most probable number per gram of soil (logMPN·g^−1^) and the others as the logarithm of colony-forming units per gram of soil (logCFU·g^−1^).

### 4.4. Enzyme Activity

Dehydrogenase activity was measured by the reduction of 2,3,5-triphenyltetrazolium chloride (TTC) into triphenyl formazan (TPF), according to Casida et al. [49]. The soil was incubated with an artificial electron acceptor, the colorless and water-soluble TTC, which is reduced due to the action of dehydrogenases to the red-colored TPF, which is insoluble in water. After the 24 h incubation, the TPF was extracted from the soil with methanol and determined by spectrophotometry (λ = 485 nm). The arylsulfatase, β-glucosidase and phosphatase activity were measured according to ISO 20130:2018 [50] in 96-well microplates. After the incubation time indicated for each enzyme, their respective substrates (potassium ρ-nitrophenyl-sulfate, ρ-nitrophenyl-β-d-glucopyranoside and ρ-nitrophenyl-phosphate) were hydrolyzed into a yellow-colored ρ-nitrophenol and determined by spectrophotometry (λ = 405 nm).

### 4.5. Photosynthetic Parameters

The photosynthetic rate (A), stomatal conductance to water vapor (gs), electron transfer rate (ETR) and chlorophyll fluorescence parameters (photochemical quenching, qP) were measured in four leaves of all the replicates for each treatment using a leaf chamber fluorometer (LI-COR 6400-40, LI-COR and Lincoln, NE, USA). The total chlorophyll content was estimated in vivo using a portable SPAD meter (CL01 Chlorophyll content system, Hansatech Instruments, Pentney, King’s Lynn, United Kingdom). The maximum quantum efficiency of PSII was measured with a Pocket-PEA (Plant Efficiency Analyzer, Hansatech Instruments, Pentney King’s Lynn, United Kingdom), Fv/Fm ratio was calculated using (Fm − F0)/Fm, where Fm is maximal fluorescence yield of the dark-adapted state and F0 is the minimum fluorescence yield [51]. All measurements were taken between 9 and 12 h in the morning.

### 4.6. Statistical Analysis

The experimental design was a complete randomized block with four replicates. The treatments were in factorial combination and consisted of two factors: plant type (with 4 levels) and soil disturbance (with 2 levels). ANOVA was performed based on the two factors using a generalized linear model, and Tukey’s test at 5% level was used to compare the means. The data were also analyzed using the Pearson correlation test, and a linear regression was performed in the results that showed moderate and strong Pearson correlation coefficients. The statistical analysis was completed using the software Minitab 21^®^ [52].

## 5. Conclusions

The results suggest that wheat growth and its functional microbiome is affected by both treatments, antecedent plant mycotrophy and imposed soil disturbance. Wheat that grew after highly mycotrophic plants (*O. compressus* and *L. rigidum*) in undisturbed soil presented higher AMF colonization and chlorophyll content, greatly improved photosynthetic rates and a 2.5-fold increase in shoot dry weight, despite the Mn toxicity. However, Mn oxidizers do not seem to have a decisive role in the detoxifying mechanism. Under these circumstances, the soil exhibited higher values of dehydrogenase activity, indicating the importance of the intact ERM as an active niche for microorganisms’ survival and maintenance of the biological activity. However, the generalist culture media used for the quantification of major microbial groups (total bacteria, phosphorus solubilizers, ammonifiers and sulfur oxidizers) did not allow the identification of differences imposed by the treatments, which was possibly due to the presence of a great proportion of unculturable or recalcitrant organisms. Wheat root colonization by AMF was highly correlated with soil basal respiration and almost all parameters of photosynthetic activity. 

Mechanisms triggered in the plant by an early mycorrhization seem to have a protective effect on the photosynthetic system. Certainly not only the AMF but also the microbiome they shape could be involved in the stress alleviation. The results presented highlight the importance of AMF in crop bio-protection strategies and its influence on the remaining soil microbiome. A further phylogenetic analysis of the microbiome changes over the studied treatments is foreseen together with the studies on other parameters of soil functional profiling, including additional time samplings. Unraveling the biological changes induced by naturally assembled AMF consortiums under toxic Mn levels could help the decision of the appropriated soil management such as the choice of crop sequence or the tillage techniques used. 

## Figures and Tables

**Figure 1 plants-11-02598-f001:**
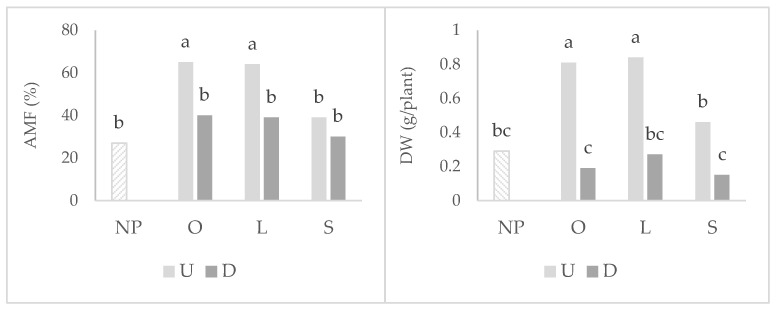
Graphs of interactions between previous plant type and soil disturbance in wheat root colonization by arbuscular mycorrhizal fungi (AMF) and shoot dry weight (DW). NP: wheat that grew with no previous plants; O: wheat that grew after *O. compressus*; L: wheat that grew after *L. rigidum*; S: wheat that grew after *S. gallica*; U: soil undisturbed and D: soil disturbed. Values sharing different letters indicate significant differences between treatments at the 5% level (Tukey’s *t*-test).

**Figure 2 plants-11-02598-f002:**
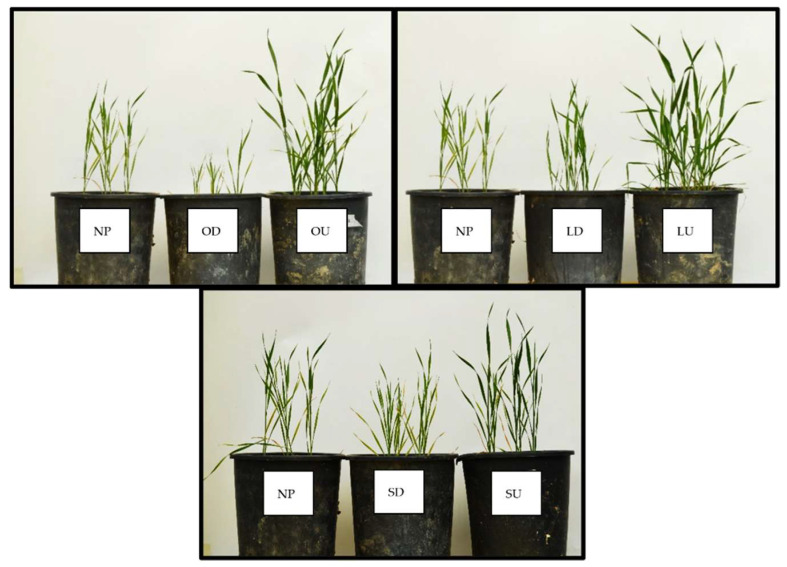
Comparison of wheat growth 21 days after planting. OD: after *O. compressus* under disturbed treatment; OU: after *O. compressus* under undisturbed treatment; LD: after *L. rigidum* under disturbed treatment; LU: after *L. rigidum* under undisturbed treatment.

**Figure 3 plants-11-02598-f003:**
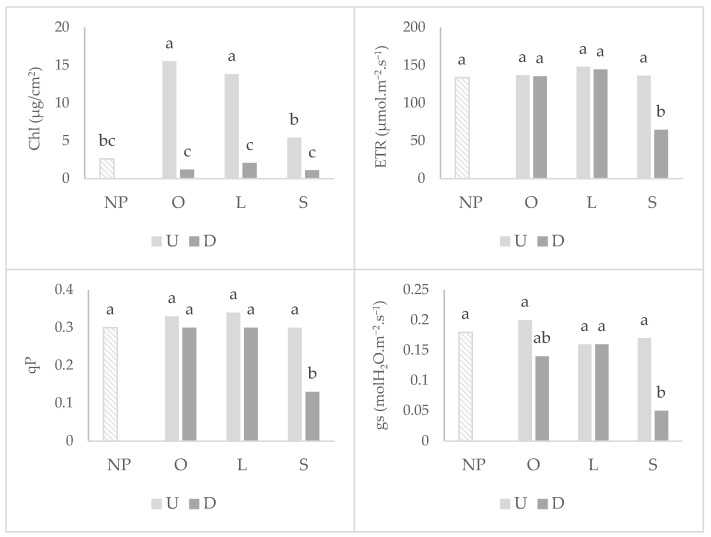
Graphs of interactions between previous plant type and soil disturbance in wheat photosynthetic parameters: chlorophyll content (Chl), electron transfer rate (ETR), photochemical quenching (qP) and stomatal conductance to water vapor (gs). NP: wheat that grew with no previous plants; O: wheat that grew after *O. compressus*; L: wheat that grew after *L. rigidum*; S: wheat that grew after *S. gallica*; U: soil undisturbed and D: soil disturbed. Values sharing different letters indicate significant differences between treatments at the 5% level (Tukey’s *t*-test).

**Figure 4 plants-11-02598-f004:**
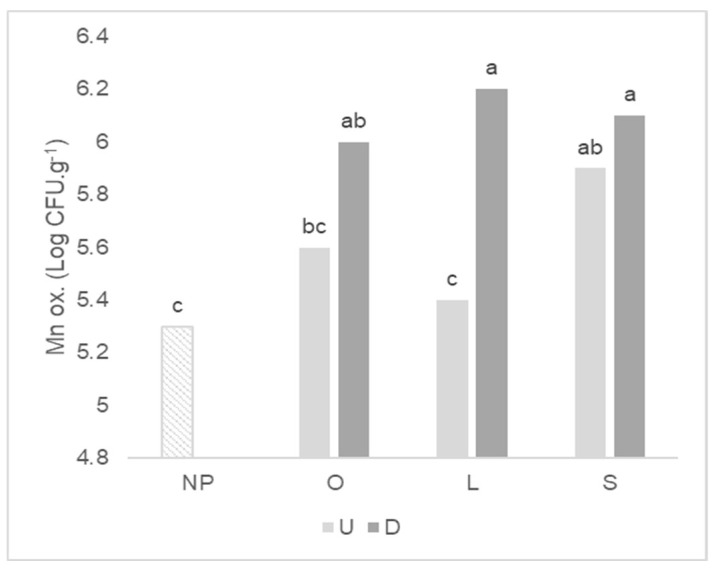
Graph of interactions between previous plant type and soil disturbance in Mn oxidizers (Mn ox.) count after wheat growth. NP: wheat that grew with no previous plants; O: wheat that grew after *O. compressus*; L: wheat that grew after *L. rigidum*; S: wheat that grew after *S. gallica*; U: soil undisturbed and D: soil disturbed. Values sharing different letters indicate significant differences between treatments at the 5% level (Tukey’s *t*-test).

**Figure 5 plants-11-02598-f005:**
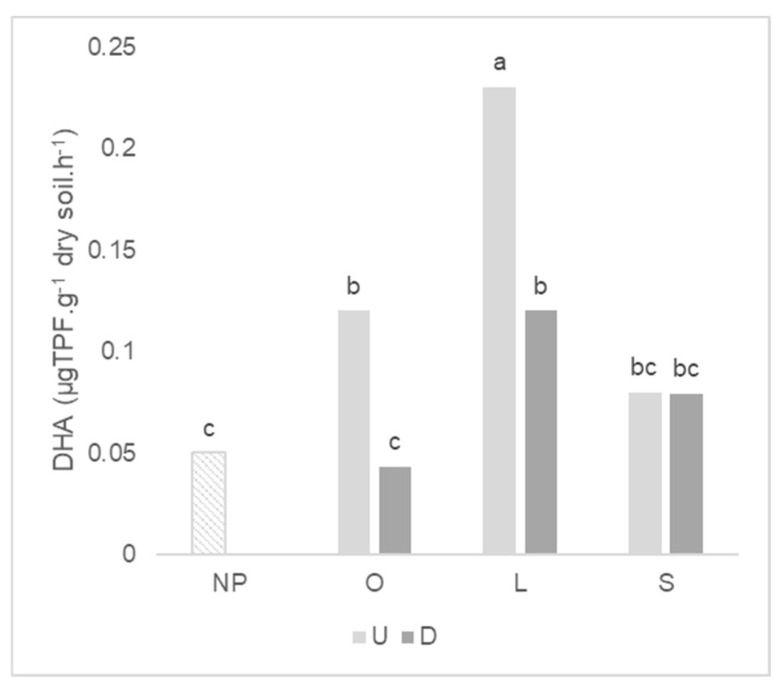
Graph of interactions between previous plant type and soil disturbance in dehydrogenase (DHA) activity after wheat growth. NP: wheat that grew with no previous plants; O: wheat that grew after *O. compressus*; L: wheat that grew after *L. rigidum*; S: wheat that grew after *S. gallica*; U: soil undisturbed and D: soil disturbed. Values sharing different letters indicate significant differences between treatments at the 5% level (Tukey’s *t*-test).

**Figure 6 plants-11-02598-f006:**
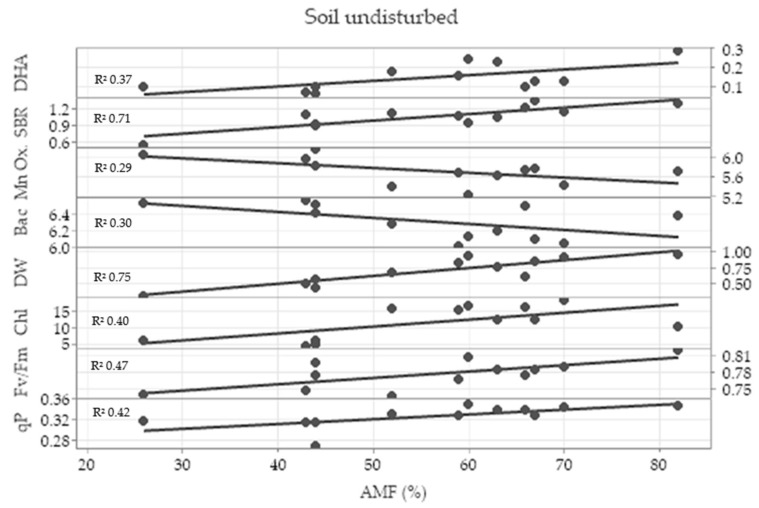
Linear regression analysis between wheat root colonization by arbuscular mycorrhizal fungi (AMF) and dehydrogenase activity (DHA), soil basal respiration (SBR), total bacteria (Bac), Mn oxidizers (Mn Ox.), shoot dry weight (DW), chlorophyll content (Chl), maximum quantum efficiency of photosystem II (Fv/Fm) and photochemical quenching (qP) in the undisturbed treatment with the coefficient of linear regression (R^2^).

**Figure 7 plants-11-02598-f007:**
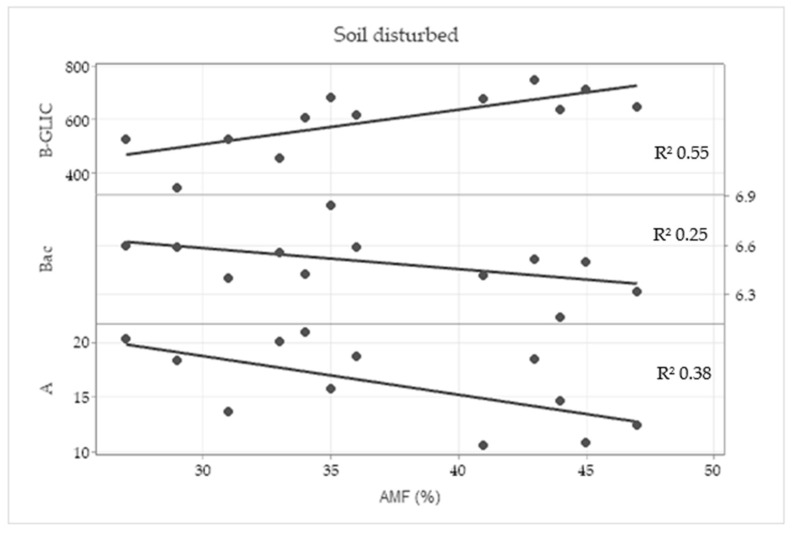
Linear regression analysis between wheat root colonization by arbuscular mycorrhizal fungi (AMF) and β-glucosidase activity (B-GLIC), total bacteria (Bac) and photosynthetic rate (A) in the disturbed treatment with the coefficient of linear regression (R^2^).

**Table 1 plants-11-02598-t001:** Effect of previous type of plant and soil disturbance on wheat photosynthetic parameters: maximum quantum efficiency of PSII (Fv/Fm) and photosynthetic rate (A).

	Fv/Fm (µg/cm^2^)				A (µmol CO_2_ m^−2^ s^−1^)			
	Soil Disturbance		Mean Plant		Soil Disturbance		Mean Plant	
Plant Type	U	D			U	D		
*O. compressus*	0.77	0.77	0.77	A	16.61	13.52	15.06	AB
*L. rigidum*	0.78	0.77	0.77	A	16.44	16.33	16.39	A
*S. gallica*	0.76	0.74	0.75	A	17.48	18.97	18.22	A
No Plant	0.7		0.70	B	0.70		0.70	B
**Mean soil**	0.74	0.75			15.57	15.14		

Values sharing different letters indicate significant differences between treatments at the 5% level (Tukey’s *t*-test). U: undisturbed treatment and D: disturbed treatment.

**Table 2 plants-11-02598-t002:** Effect of previous type of plant and soil disturbance on soil basal respiration (SBR), microbial biomass carbon (C-Mic) and metabolic quotient (qCO_2_) after wheat growth.

	SBR				C-Mic			qCO_2_			
	(mg CO_2_ Kg^−1^ Soil h^−1^)				(mg C Kg^−1^ Soil)			(mg CO_2_ mg^−1^ C-mic h^−1^) ×10^−3^			
	Soil Disturbance		Mean Plant		Soil Disturbance		Mean Plant	Soil Disturbance		Mean Plant	
Plant Type	U	D			U	D		U	D		
*O. compressus*	1.19	1.13	1.16	A	137.73	106.48	122.10	8.8	10.6	9.7	A
*L. rigidum*	1.09	1.08	1.09	AB	113.42	114.58	114.00	9.9	9.7	9.8	A
*S. gallica*	0.86	1.04	0.95	B	129.63	118.05	123.84	6.9	9.3	8.1	AB
No Plant	0.54		0.54	C	99.53		99.53	5.5		5.5	B
**Mean soil**	0.92	0.95			120.08	109.66		7.8	8.8		

Values sharing different letters indicate significant differences between treatments at the 5% level (Tukey’s *t*-test). U: undisturbed treatment and D: disturbed treatment.

**Table 3 plants-11-02598-t003:** Effect of previous type of plant and soil disturbance on soil microbial functional group count of total bacteria, phosphorus (P) solubilizers, ammonifiers and sulfur (S) oxidizers after wheat growth.

	Total Bacteria						P Solubilizers				Ammonifiers					S Oxidizers				
	(Log CFU g^−1^)										(Log MPN g^−1^)									
	Soil Disturbance				Mean Plant		Soil Disturbance		Mean Plant		Soil Disturbance				Mean Plant	Soil Disturbance				Mean Plant
Plant Type	U		D				U	D			U		D			U		D		
*O. compressus*	6.16		6.32		6.24	B	5.49	5.31	5.40	AB	7.32		8.91		8.11	3.14		4.39		3.77
*L. rigidum*	6.24		6.61		6.42	AB	5.52	5.95	5.74	A	6.93		8.24		7.59	3.44		4.28		3.86
*S. gallica*	6.50		6.54		6.52	A	5.16	5.37	5.26	B	7.44		8.04		7.74	3.65		4.23		3.94
No Plant	6.25				6.25	B	5.66		5.66	AB	7.30				7.30	3.63				3.63
**Mean soil**	6.29	B	6.43	A			5.46	5.57			7.25	B	8.12	A		3.47	B	4.13	A	

Values sharing different letters indicate significant differences between treatments at the 5% level (Tukey’s *t*-test). U: undisturbed treatment and D: disturbed treatment.

**Table 4 plants-11-02598-t004:** Effect of previous type of plant and soil disturbance on soil enzymatic activities of arylsulfatase, phosphatase and β-glucosidase after wheat growth.

	Arylsulfatase				Phosphatase				β-Glucosidase					
	(nmol ρ-Nitrophenol g^−1^ Dry Soil h^−1^)													
	Soil Disturbance		Mean Plant		Soil Disturbance		Mean Plant		Soil Disturbance				Mean Plant	
Plant Type	U	D			U	D			U		D			
*O. compressus*	37.50	41.22	39.36	AB	2091.81	2250.25	2171.03	B	499.09		621.40		560.24	AB
*L. rigidum*	42.86	41.61	42.24	A	2255.14	2391.39	2323.27	AB	538.91		687.76		613.34	A
*S. gallica*	37.66	36.84	37.25	AB	2430.72	2600.24	2515.48	A	476.75		483.34		480.04	B
No Plant	33.56		33.56	B	1908.00		1908.00	C	548.69				548.69	AB
**Mean soil**	37.90	38.31			2171.42	2287.47			515.86	B	585.30	A		

Values sharing different letters indicate significant differences between treatments at the 5% level (Tukey’s *t*-test). U: undisturbed treatment and D: disturbed treatment.

## Data Availability

Not applicable.

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
