# Peer review of "Influence of Intact Mycelium of Arbuscular Mycorrhizal Fungi on Soil Microbiome Functional Profile in Wheat under Mn Stress"

_plants, 2022, doi:10.3390/plants11192598_

Round 1

Reviewer 1 Report

This publication presents interesting results related to the changes in soil functional profile and wheat growth performances induced by antecedent plant mycotrophy and soil disturbance under Mn stress. The adopted approach in this study was very interesting since it could reduce Mn toxicity and boost soil microbial activity.

The manuscript was well introduced, and the authors adopted convincing methods with a discussion of the different obtained results. However, the manuscript needs substantial revisions to be suitable for publication in Plants.

General comments

- Comment 1: The English of this manuscript needs substantial improvements.

- Comment 2: There are missing spaces, incorrect punctuation, and typing errors that show that the authors did not have a final reading of the article before it was submitted.

- Comment 3: The scientific names of species should be written in italic in the whole manuscript.

- Comment 4: It would be interesting if you measured Mn content in soil (before and after the experiment) and wheat samples to conclude the effect of AMF on mitigating Mn stress.

Other comments

- Title: please change AMF to Arbuscular mycorrhizae.

- Abstract

L20-23: please rewrite.

L25: please change “measures” to “values”.

L23-26: please specify compared to what?

- Introduction

L52: please provide the significance of AMF.

L62: please change “in” to “by”.

L55-58: please change “to an increase in 1.5-fold in wheat” to “to a 1.5-fold increase in wheat”.

L78: please add SOM after soil organic matter.

- Results

L149: please change “previous type of plant” to “previous type of mycotrophic plant”. Please do the same for all the illustrations

Table 1: please check the values of AMF colonization. They are very low as percentages !! Maybe they should be x100

L151-153: It is necessary to mention the difference between statistical upper- and lower-case letters for each illustration.

- Discussion

L225: Please place the heading “Discussion” after Table 5 footnote.

L243-245: You cannot rely on this result since it was insignificant.

L277-278: It is too easy to say and we did not have this information in Figure 1.

Please discuss the results related to soil microbial activity.

How do you explain the greatest value of phosphatase recorded in the soil where wheat grew after the non-mycotrophic plant? Please include it in the discussion.

Please discuss the result related to phosphorus solubilizers, ammonifiers, and sulfur oxidizers by showing how they can be involved in the mitigation of Mn toxicity.

- M&M

L322: please add a space between the value and the unit (please do the same throughout the manuscript)

L334: It is just 24 pots instead of 32?

- Conclusion

The conclusion should be reduced to the main findings.

Reviewer 2 Report

The manuscript submitted by Conceicao et al. was experimentally overall well done and the experimental set-up is well described but the presentation of the results and their description need substantial improvement.  The whole text requires substantial polishing by a native speaker familiar with scientific writing.

I do not agree with the conclusions and title of the manuscript which are not supported by the experimental set-up and results. The authors show differences in AMF colonization rates and changes in microbial functions but they do not show any causal relationship between the two. How can the authors rule out that other factors do not play a role in the shift e.g. soil disturbance may physico-chemical parameters such as aeration, pore size, nutrient distribution? Furthermore, the different plants used as catch crop/ pretreatment may acquire nutrients differently - and based on your results indeed the catch crop seems to be more important than the ERM/disturbance. Most of the significant differences you show depend on the plant species used first. It is ok to assume that effects are based on ERM but wording should be much more careful and precise and alternative hypothesis discussed as well.

The description of the results is very hard to follow. It is partly repetitive, first you give a general picture and then you repeat what you see. Please read more literature and adapt your style to scientific writing. In this context it would also be good to show figures and not only tables. Please also explain in the table legend that Ornthopus, Lolium etc. are the plants grown first in the pots before planting wheat.

I do not fully understand why you have used only a single type of pots for "no plants". Sure there is no plant that could have built up a network but you have watered the plants and kept the pots in the glasshouse likely changing the microbial community already during the phase without wheat.

Please delete the term "Developer plants" from your manuscript. It does not help the reader to understand the manuscript and it is misleading.

Round 2

Reviewer 2 Report

Many thanks for the revised version of the manuscript. Unfortunately, there are still quite a few points that have not been addressed by the authors.

Looking at tables 3 and 4 the authors show significant differences depending on the plant grown before wheat but there are no significant differences (except of Mn oxidizers) between the treatment (disturbed/undisturbed). However, AMF colonization rates of plants in the disturbed treatment are the same independent of the antecedent plant (table 1). Based on these results I do not understand why AMF shapes the soil microbiome profile in wheat. Hence, I agree with the authors that the plant previously grown in the pot has an influence on wheat growth, photosynthesis and soil/plant nutrient content but the effect of AMF on the microbiome could not be shown in this set-up. Can you correlate AMF colonization rates with bacterial counts? What about a correlation between colonization rates and photosynthesis parameters or simply wheat biomass?

I still think it would be useful to visualize some of the results to get a clearer picture of the changes due to the different treatments. Furthermore, I still recommend to ask a native speaker to polish the manuscript. The manuscript reads a bit better but still contains a number of mistakes and phrasings that need substantial improvement.

Round 3

Reviewer 2 Report

Many thanks for the revised version of the manuscript. I acknowledge that the authors revised the text and introduced two figures. The text ready much better than before but it still contains a large number of mistakes. In this context the authors should also re-visit the format of the numbers in the graphs and text of subchapter 2.5 ( comma vs. dot). Overall, I ask the authors again to seek advice from a native English speaker. It is fine to get scientific comments first and not to seek help right from the beginning, but the second reviewer and I pointed it out twice that the text needs to be polished so that it is really annoying that the manuscript is still full of mistakes. I also disagree that it is a good scientific standard to show tables only. It is very helpful to see the complete data in the supplementals but I also expect the authors to highlight some of the important findings by showing these data as graphs. Overall the scientific style of the manuscript is definitely below the level of PLANTS.

In this context, please be precise in the re-written subchapter 2.4. Which enzymatic activity was significantly affected and greatest in undisturbed soil after growth of L. rigidum?

What is an AMF rate?

What is a relevant increase? A significant increase or relevant for what?

The last new paragraph of the discussion needs to be integrated into the discussion and not presented as an addendum.
